# Effect of H_2_O_2_ Bleaching Treatment on the Properties of Finished Transparent Wood

**DOI:** 10.3390/polym11050776

**Published:** 2019-05-01

**Authors:** Yan Wu, Jiamin Wu, Feng Yang, Caiyun Tang, Qiongtao Huang

**Affiliations:** 1College of Furnishings and Industrial Design, Nanjing Forestry University, Nanjing 210037, China; wujiamin012@163.com (J.W.); tangcaiyun012@163.com (C.T.); 2Fashion Accessory Art and Engineering College, Beijing Institute of Fashion Technology, Beijing 100029, China; 3Department of Research and Development Center, Yihua Lifestyle Technology Co., Ltd., Shantou 515834, China; huangqt@yihua.com

**Keywords:** transparent wood, chemical composition, H_2_O_2_ bleaching treatment, physicochemical properties

## Abstract

Transparent wood samples were fabricated from an environmentally-friendly hydrogen peroxide (H_2_O_2_) bleached basswood (*Tilia*) template using polymer impregnation. The wood samples were bleached separately for 30, 60, 90, 120 and 150 min to evaluate the effects on the changes of the chemical composition and properties of finished transparent wood. Experimental results showed decreases in cellulose, hemicellulose, and lignin content with an increasing bleaching time and while decreasing each component to a unique extent. Fourier transform infrared spectroscopy (FT-IR) and scanning electron microscope (SEM) analysis indicated that the wood cell micro-structures were maintained during H_2_O_2_ bleaching treatment. This allowed for successful impregnation of polymer into the bleached wood template and strong transparent wood products. The transparent wood possessed a maximum optical transmittance up to 44% at 800 nm with 150 min bleaching time. Moreover, the transparent wood displayed a maximum tensile strength up to 165.1 ± 1.5 MPa with 90 min bleaching time. The elastic modulus (*E_r_*) and hardness (*H*) of the transparent wood samples were lowered along with the increase of H_2_O_2_ bleaching treatment time. In addition, the transparent wood with 30 min bleaching time exhibited the highest *E_r_* and *H* values of 20.4 GPa and 0.45 GPa, respectively. This findings may provide one way to choose optimum degrees of H_2_O_2_ bleaching treatment for transparent wood fabrication, to fit the physicochemical properties of finished transparent wood.

## 1. Introduction

Wood-based materials are widely used in our living environments, such as for housing construction, interior decoration, and furniture manufacturing, etc. [1]. As a novel wood-based material, transparent wood has attracted increasing research interest due to its renewable raw material heritage, outstanding optical transmittance and haze, strong durability as well as its mechanical properties, and finally, its low thermal conductivity [2,3]. The multifunctional transparent wood material not only possessed similar mechanical characteristics to wood in engineering applications, but its unique optical properties can also be beneficially utilized across new fields to expand the applications of the wood. Some examples of these applications include the use of transparent wood as an illuminable structural medium, as a planar light source in luminescent buildings, or as a component of energy efficient smart building, and transparent wood materials are also suitable for application in electronic devices such as conductive substrate, etc. [4,5,6,7,8].

Lignin plays a key role in the visual properties of wood and wood-derived materials, serving as the primary contributor to the opaque color of native wood [9]. To make transparent wood, a crucial step involves beginning with a delignified wood. The superstructure of wood after delignification is then filled with a polymer (or combination of polymers) with a desired refractive index. The finished visual properties of transparent wood is therefore primarily driven by the degree of delignification, as well as the void-filling polymer system chosen. 

Recent literature on this specific topic is full of various delignification approaches used to render transparent wood templates (and eventually transparent wood). Fink (1992) developed transparent wood materials based on placing the wood samples for 1–2 days of submersion in a 5% aqueous solution of sodium hypochlorite to remove colored components, including lignin [10]. Li et al. (2016) reported that their delignified transparent wood templates were also prepared using submersion in sodium chlorite (NaClO_2_), which successfully lowered the lignin content from 24.9% to 2.9% [11]. Qiu et al. (2019) also used 1.5 wt% NaClO_2_ with an acetate buffer solution (pH 4.6) as a lignin removal solution to impregnate wood samples, and lignin content decreased from 23.5 ± 1.8% for the untreated wood to 1.6 ± 0.2% for the delignified wood after 8 h delignification [12]. Using a different approach, Zhu et al. (2016) reported a means of delignification using a boiling aqueous solution of NaOH and Na_2_SO_3_, to which additional H_2_O_2_ was added. In this case, lignin content in the template woods was less than 3% [13]. Liu et al. (2018) also obtained delignified wood by using a 5 g NaOH and 15 g Na_2_SO_3_ mixing 400 mL methanol (20% volume fraction) water solution to extract wood samples, then the samples were placed in the 1.5 mol/L H_2_O_2_ solution until the wood yellow color disappeared and the removal rate of lignin reached up to 99.2% [14]. Generally, there are harmful components, such as methyl mercaptan, dimethyl sulfide, and hydrogen sulfide, generated during the delignification process [15]. As can be surmised, these sorts of delignification processes are time-consuming and not necessarily environmentally friendly. In addition, the severity of such processes weakens the mechanical properties of the wood template due to excessive lignin removal. Therefore, it is imperative to optimize the wood template preparation process along the lines of achieving as much lignin retention as possible, and finished transparent wood possessed high light transmittance without significantly sacrificing mechanical properties.

The purpose of this work was to develop a better understanding of how extent of bleaching treatment in transparent wood templates relates to the properties of finished transparent wood materials. An environmentally friendly H_2_O_2_ bleaching process was adopted, which includes H_2_O_2_ as a bleaching agent and trisodium citrate dihydrate as a pH stabilizer (used in place of more harmful pH stabilizers). The effect of varying delignification time on the chemical composition and morphological, optical, and macromechanical properties of transparent wood was investigated. In addition, the micromechanical properties of the finished transparent wood were also observed using nanoindentation techniques. The goal of these analyses was to promote the utilization of transparent wood as a novel bio-based material that is both visually appealing and mechanically functional.

## 2. Experimental

### 2.1. Materials

Basswood (Tilia) with dimensions of 20 mm long × 20 mm wide × 0.4 mm thick (the depth of the lumina is as long as the length of the wood samples) and ultrapure water were supplied by Yihua Lifestyle Technology Co., Ltd., (Guangdong, China). Trisodium citrate dihydrate was purchased from Sinopharm Chemical Reagent Co., Ltd., (Shanghai, China). Sodium hydroxide (NaOH), Hydrogen peroxide (H_2_O_2_, 30% solution) and methyl methacrylate (MMA) were provided by Xilong Scientific Co., Ltd., (Guangdong, China). Ethanol was supplied from Tianjin Fuyu Fine Chemical Co., Ltd., (Tianjin, China). 2,2′-Azobis (2-methylpropionitrile) (AIBN) was obtained from Tianjin Benchmark Chemical Reagent Co., Ltd., (Tianjin, China). 

### 2.2. Fabrication of Transparent Wood

Natural wood (NW) samples were first dried at 103 °C for 24 h prior to further bleaching treatment. The bleaching solution was prepared through mixing 6 wt % H_2_O_2_, 1 wt % trisodium citrate dihydrate, 1 wt % NaOH, and 92 wt % ultrapure water [16]. Afterwards, the bleached wood (BW) samples were prepared from the dried NW by treating it with this bleaching solution at 60 °C for different bleaching times. The Alkaline H_2_O_2_ bleaching treatment is an environmentally friendly method where the chromophore structures in lignin were removed or selectively reacted and most of lignin was preserved [17]. Because trisodium citrate dihydrate is a safe, non-toxic and biodegradable reagent, it has good PH regulation and buffering performance and could be used as a stabilizer in the process of H_2_O_2_ bleaching [18]. The bleached wood samples produced were labeled as BW-30, BW-60, BW-90, BW-120, BW-150 samples. These samples correspond to the BW samples produced from NW samples treated for 30, 60, 90, 120 and 150 min, respectively. After treatment, the BW samples were thoroughly washed with ultrapure water and then suspended in ethanol prior to preparation of transparent wood. To begin transparent wood production, pure MMA monomer was uniformly mixed with AIBN initiator (0.5 wt % solution) and pre-polymerized at 75 °C for 15 min [11]. After the designated pre-polymerization time, the impregnation solution was cooled to room temperature. The BW samples were immersed in cooled prepolymerized solution under vacuum for 30 min. Later, the vacuum pump was turned off and the polymer solution was allowed to continue to fill the wood templates for an additional 1 h. Finally, the polymer-infiltrated wood was sandwiched between two pieces of glass and transparent wood (TW) samples were obtained by heated at 70 °C for 5 h. As far as sample labeling is concerned, TW-1, TW-2, TW-3, TW-4 and TW-5 were (respectively) produced from the BW-30, BW-60, BW-90, BW-120 and BW-150 templates. An illustration of the entire preparation process and the finished samples are provided as Figure 1.

### 2.3. Chemical Composition Content Analysis

The cellulose, hemicellulose and lignin contents (including acid-insoluble lignin plus acid-soluble lignin) of NW and BW samples were tested by the Laboratory Analytical Procedure (LAP) written by the National Renewable Energy Laboratory (Determination of Structural Carbohydrates and Lignin in Biomass). The main methods were as follows [19]: First, samples were processed into 20–80 meshes of wood powder and dried at 105 °C for 24 h. Then 0.3 ± 0.01 g dried wood powder were put into the hydrolytic flask. The 3.00 ± 0.01 mL 72 wt % concentrated sulfuric acid was added to the hydrolytic flask, and all wood powder samples were infiltrated at the same time. The hydrolysate flasks were all covered and placed in water bath at 30 °C for 1 h. Then the 84.00 mL ± 0.04 mL water was added in the flask. Then flasks were placed into the sterilizer at 121 °C for 1 h and opened after cooling. The hydrolyzed samples were filtered by a constant weight G3 funnel, and 50 mL filtrate was retained for the determination of acid-soluble lignin (measured within 6 h) and sugar concentration. The filtered residue was rinsed with hot deionized water to neutral, and then placed in an oven at 105 °C until constant weight, and the weight was recorded. After constant weight, it was transferred to the muffle furnace and dried at 575 ± 25 °C for 24 ± 6 h. Later, the G3 funnel was taken out and the weight was recorded. The filtrate was diluted with the corresponding multiple, and the UV absorption value was determined by UV spectrophotometer at 205 nm. The filtrate was diluted by a certain multiple, and then the sugar content was analyzed by High performance liquid chromatography (HPLC).

In addition, the content and removing rate of the chemical composition in Table 1 and Figure 2 can be determined by Formulas (1) and (2), respectively [20].
(1)$$\mathrm{Content}\;(\%)=\frac{M_{B}}{M_{W}}\times 100\%$$
(2)$$\mathrm{Removal}\;\mathrm{percentage}(\%)=\frac{M_{\mathrm{NW}}-M_{\mathrm{BW}}}{M_{\mathrm{NW}}}\times 100\%$$
where M_B_ is the mass of each chemical composition content in samples, M_W_ is the total mass of the samples, M_NW_ is the mass of each chemical composition content in NW samples, and M_BW_ is the mass of each chemical composition content in BW samples.

### 2.4. Fourier Transform Infrared Analysis

The attenuated total reflection (ATR) Fourier transform infrared spectroscopy (FT-IR) spectra of NW, BW and TW samples were analyzed using a VERTEX 80 V spectrometer. Spectra were collected over the range from 400 cm^−1^ to 4000 cm^−1^ by 16 scans at a resolution of 4 cm^−1^. The dimensions of samples were 20 mm long × 20 mm wide × 0.4 mm thick and all samples were dried before analysis. 

### 2.5. Scanning Electron Microscopy

The dried NW, BW and TW samples were coated with gold particles, and then observed with a FEI Quanta 200 scanning electron microscope (SEM) at an accelerating voltage of 20 kV. The cross sections of samples that perpendicular to the direction of the wood fiber alignment were observed.

### 2.6. Optical Properties

The optical transmittance of NW and TW samples were collected by a *Shanghai*
*youke* UV1900PC spectrophotometer with a wavelength ranging from 350 nm to 800 nm.

### 2.7. Mechanical Properties

The tensile strength of NW and TW samples were performed in a SANS-CMT6104 electromechanical universal testing machine. The tensile speed was set at 2 mm/min. The dimensions of samples were 20 mm long × 20 mm wide × 0.4 mm thick and stretched in the direction of the fiber alignment. 

### 2.8. Nanoindentation 

The nanoindentation test method, the reduced elastic modulus (MOE) (*E*_r_) and hardness (*H*) measurements upon TW samples, and calculation methods (Formulas (3) and (4)) were all performed according to our previously described research methods [21,22]. The nanoindentation test performed by a Hysitron TriboIndenter system (Hysitron, Inc., Eden Prairie, MN, USA) with scanning probe microscope (SPM) and a three-sided Berkovich type indenter. About 30 valid indents on the wood cell wall were analyzed when the peak load was 400 µN based on a load-controlled mode (the loading time was 5 s, the holding time was 5 s and the unloading time was 5 s). The calculation methods of reduced elastic modulus (MOE) (*E*_r_) and hardness (*H*) were as follows: Formulas (3) and (4), respectively.
(3)$$E_{r}=\frac{\sqrt{\pi }}{2\beta }\frac{S}{\sqrt{A}}$$
(4)$$H=P_{\max}/A$$
where S is the initial unloading stiffness and β is a correction factor correlated to indenter geometry (β = 1.034), P_max_ is the peak load, and A is the projected contact area of the indents at peak load.

## 3. Results and Discussion

### 3.1. Chemical Composition Content Analysis

The NW samples was treated by H_2_O_2_ bleaching solution and the changes of wood cell wall components as shown in Table 1 and Figure 2. The key components of wood cell wall are linear polysaccharide cellulose, heterogeneous hemicellulose, structurally variable lignin, linkage with each other via hydrogen bond network (between cellulose and hemicellulose) and covalent linkage (between lignin and hemicellulose) [23,24]. Natural lignin is a three-dimensional amorphous polymer with a dark color comprising three types of lignin units, termed syringyl units (S), guaiacyl units (G), and p-hydroxyphenyl units (H) [25]. As can be seen in Table 1 and Figure 2, increasing H_2_O_2_ bleaching times resulted in increasing removal rates of cellulose, hemicellulose, and lignin. It was found that lignin was the most affected by H_2_O_2_ bleaching treatment. Regarding lignin removal, the lignin content decreased from 24.3% (NW) to 14.9% (BW-150), translating to a 38.7% delignification in BW-150 compared to NW. A slight decline of cellulose and hemicellulose content took place, decreasing from 48.3% (NW) and 17.2% (NW) to 43.5% (BW-150) and 15.7% (BW-150), respectively. In addition, neither cellulose nor hemicellulose had a removal percentage of more than 10%. One interesting observation was found, in that the removal of cellulose, hemicellulose, and lignin was greatest in the first 30 min of bleaching time. As time was prolonged, the additional removal rates decreased. However, a slightly linear relationship can be visually assessed for lignin removal and the prolonged bleaching times of BW samples. The chromophore structures of lignin were destroyed by oxidizing the carbonyl structure and quinoid structure of lignin side chain during H_2_O_2_ bleaching [26]. Furthermore, H_2_O_2_ may react with the benzoquinone structure of lignin to change the solubility of lignin, or react with the side chain carbonyl and carbon-carbon double bond of lignin to further oxidative degradation, leading to lignin removal [27]. Also, during the H_2_O_2_ bleaching treatment with the existence of alkali, the carbohydrate may be oxidized and degradation of carbohydrate would have occurred [28,29]. The effect of the H_2_O_2_ bleaching treatment on the wood samples was as expected.

### 3.2. Fourier Transform Infrared Analysis

Figure 3 shows the FTIR spectra of NW compared against each BW and TW sample. In Figure 3a, characteristic absorption bands of NW samples are marked in the regions of 3330 cm^−1^ (O–H stretch), 2900 cm^−1^ (C–H stretch), 1730 cm^−1^ (C=O stretch in hemicellulose), 1590 cm^−1^ and 1500 cm^−1^ (C=C stretching vibration from lignin), 1370 cm^-1^ (C–H deformation vibration in cellulose and hemicellulose), 1240 cm^−1^ (C–O stretching in lignin and hemicellulose) in Figure 3a [30,31]. The band at 1500 cm^−1^ remained almost unchanged after bleaching treatment as the aromatic lignin had resistance to the hydrogen peroxide treatment [32]. Also after bleaching treatment, the bands at 1240 cm^−1^ and 1370 cm^−1^ in BW samples were shown to be smaller than what is observed in NW samples. In addition, the band at 1730 cm^−1^ (attributed to acetyl groups in hemicelluloses) also weakened and eventually disappeared as the bleaching time increased. The observed changes to the bands were consistent with chemical composition content changes in Table 1, where removal of all three components was observed at increasing treatment times [33]. The spectrum of transparent wood (Figure 3b) showed peaks at 2990 cm^−1^ and 2950 cm^−1^ (C−H stretching of the methyl group), 1720 cm^−1^ (C=O stretching of the ester carbonyl), 1190 cm^−1^ and 1150 cm^−1^ (C−O−C stretching of the ester group), all of which were characteristic of the impregnation polymer PMMA [34]. This observation confirms successful impregnation of MMA solution into the wood template.

### 3.3. Scanning Electron Microscopy

As intended, both H_2_O_2_ bleaching treatment and polymer impregnation resulted in structural changes on the wood surfaces. SEM images acquired of NW, BW and TW samples are each shown in Figure 4 to provide a qualitative assessment of the microstructures of each sample. In addition, simple photographs of NW, BW (BW-150) and TW (TW-5) samples are also shown in Figure 4. As can be seen, the BW (BW-150) sample (Figure 4b) was colorless relative to the NW sample (Figure 4a). This is expected, as we have already established that bleaching removed a fair quantity of color-providing lignin. Speaking to the favorable transparent properties of the TW, the grid lines underneath the TW (TW-5) sample can be clearly seen in Figure 4c. This transparency is also an additional demonstration of successful polymer impregnation. Moving on to microstructural considerations, the microscale pores and their aligned channels can be observed in the SEM images of the BW samples (Figure 4b). This shows that such pores were preserved (comparing Figure 4a,b) in spite of the bleaching treatment while removing color. The main observable difference between the NW and BW microstructures includes the changes to the structure of the cell walls. In NW, the cell wall appears less uniform, while the uniformity increases in the BW samples despite the apparent pore diameters decreasing. We attribute this difference to removal of both lignin and hemicellulose, biopolymers which are often collectively thought of as binding agents in plant cell wells. By chemically removing a healthy quantity of these binding agents, we are forcing the micromolecular structure to adopt more uniform conformations [35]. As such, we have demonstrated that the bleaching treatment is not severe enough to collapse the lumens of the plant cells, but is selective enough to reduce the visual color of the material itself. Figure 4c shows the SEM images of the TW samples. The displayed TW (TW-5) micrographs show that almost of the cell lumens were successfully filled with the polymer. Furthermore, effective bonding between the interfaces of the wood template surfaces and PMMA polymer and the wood cell walls can be observed. This observation is consistent with similar images and observations reported by Wang et al [36].

### 3.4. Optical Properties

Figure 5 shows the optical transmittance of the TW samples with the NW transmittance shown as a control. As a control, the NW sample (24.3% lignin content) exhibited almost 0% optical transmittance due to both light absorption by lignin and light scattering by the porous wood structure [37]. In contrast, it can be seen that TW-5 samples exhibited optical transmittance as high as 44%. The optical transmittance was also found to slightly increase as a function of the template’s bleaching time. This finding is congruent to our previous studies, which showed that increasing lignin removal in transparent wood templates contributes to an increase of an optical transmittance value of the finished transparent wood [20]. Specifically, when the lignin content (Table 1) decreased from 19.5% (BW-30) to 14.9% (BW-150), the optical transmittance slightly increased from 38% (TW-1) to 44% (TW-5) at 800 nm. Beyond lignin removal, each TW sample’s transmittance could also be influenced by the viscosity, refractive indices, and shrinkage properties of the PMMA impregnated within [38]. Finally, Figure 5 shows that the optical transmittance of each TW sample increases with measured wavelength (from 350 to 800 nm).

### 3.5. Mechanical Properties

Figure 6 shows the comparison of tensile strength for NW and a series of TW samples. The tensile strength of TW samples were related to their compositions and chemical structures [39]. All TW samples exhibited stronger tensile strength than NW samples (121.9 ± 6.12 MPa). These results demonstrated that not only was the transparency significantly increased by polymer infiltration, but that mechanical properties could also be improved compared to NW samples [40]. The longitudinally oriented cellulose nanofiber structure and the interaction between wood cellulose nanofibers and PMMA were favorable to the strength of TW samples [41]. It was interesting to find that out of all the TW samples, TW-3 exhibited the greatest tensile strength (165.1 ± 1.5 MPa). An overall pattern of the TW tensile strengths increasing first and then decreasing was observed. This could be due to the wood templates at longer bleaching times being mechanically weaker than the templates produced at shorter bleaching times [42].

### 3.6. Nanoindentation

The reduced elastic modulus (MOE) (*E_r_*) and hardness (*H*) of TW cell walls were summarized in Figure 7. It is shown that as the bleaching time increased, both *E_r_* and *H* of the TW samples obtained showed a downward trend that is concurrent with the removal of lignin, hemicellulose, and cellulose. The TW made from the template with the shortest bleaching time, TW-1, samples exhibited the highest *E_r_* and *H* values of 20.4 and 0.45 GPa. In comparison, the TW-5 samples exhibited the lowest *E_r_* and *H* values of 16.4 and 0.37 GPa. With the removal of the wood cell’s structural biopolymers at extended bleaching times, the interaction between the main components of the cell wall was weakened, which destroyed the integrity of the cell wall to a certain extent and decreased the *E_r_* of the cell wall [43]. Concerning hemicellulose, its function as an interface coupling agent between cellulose and lignin was increasingly damaged at prolonged bleaching times. Therefore the removal of hemicellulose observed likely also provided significant effect on the *E_r_* of the cell wall in the finished TW materials [43]. Another metric, the cellulose microfibril angle, has also been found to be correlated to the *E_r_* of the wood cell wall [44]. However, the other property measured, *H*, is governed by the yielding of the cell wall matrix (lignin and hemicelluloses), instead of the property of the microfibrils (or their alignment) [45]. Therefore extraction of hemicellulose and migration of lignin from the matrix surrounding cellulose microfibrils creates pores in the matrix that decrease the density and hardness of the material [46]. The lignin is a dimensional phenolic polymer which imparts rigidity and hydrophobicity to biomass and has a greater effect than hemicellulose on the *H* of the cell wall [43,47]. The results of nanoindentation were not correlate with the tensile strength. This could be that *E_r_* and *H* of the TW samples were influenced by the surrounding PMMA polymer, the interface between wood and PMMA polymer, the position of obtained wood sample, and so on.

## 4. Conclusions

An environmentally friendly H_2_O_2_ bleaching treatment method was successfully used to make TW samples. FT-IR and SEM observations showed that the microstructure of the wood cells was still well preserved after bleaching treatment and the MMA was successfully impregnated into the wood template. The optical transmittance and tensile strength of TW samples produced from these templates was superior to NW samples, in spite of the severe chemical compositional changes imparted by the bleaching treatment. The transparent wood possessed a maximum optical transmittance of up to 44 % at 800 nm and displayed a maximum tensile strength of up to 165.1 ± 1.5 MPa. The *E_r_* and *H* of the cell wall templates present in the TW sample was also found to be related to the removal of chemical components, as they both decreased at increasing bleaching times. Also, they exhibited the highest *E_r_* and *H* values of 20.4 and 0.45 GPa. These findings will provide a knowledge base for the further application of transparent wood as a novel home material in various fields of interior decoration.

## Figures and Tables

**Figure 1 polymers-11-00776-f001:**
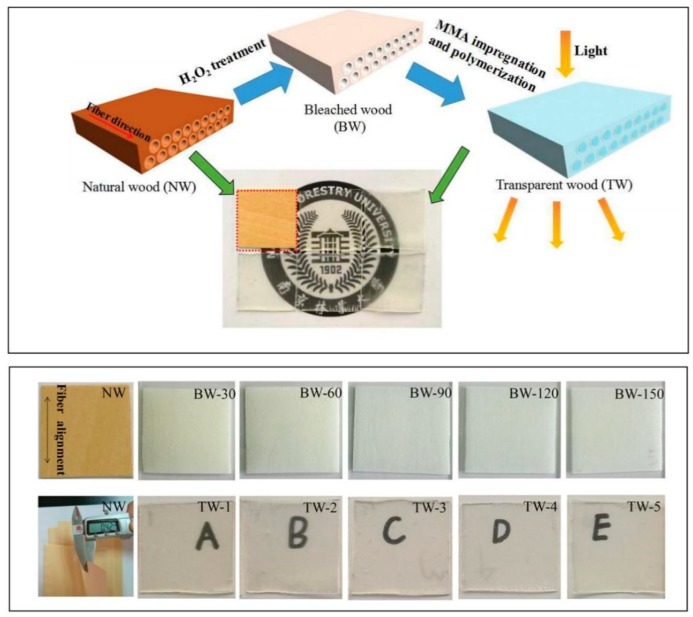
Schematic illustration of the preparation processes for transparent wood and finished samples.

**Figure 2 polymers-11-00776-f002:**
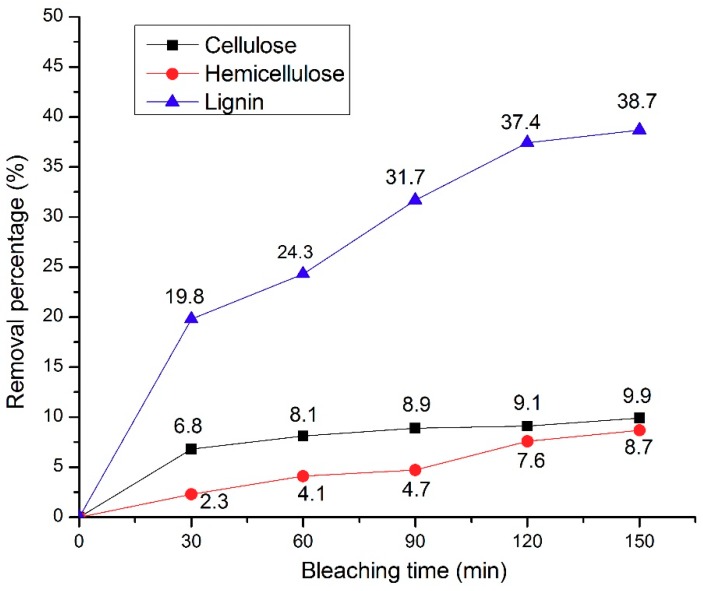
The removal percentage of cellulose, hemicellulose and lignin in wood samples at increasing H_2_O_2_ bleaching times.

**Figure 3 polymers-11-00776-f003:**
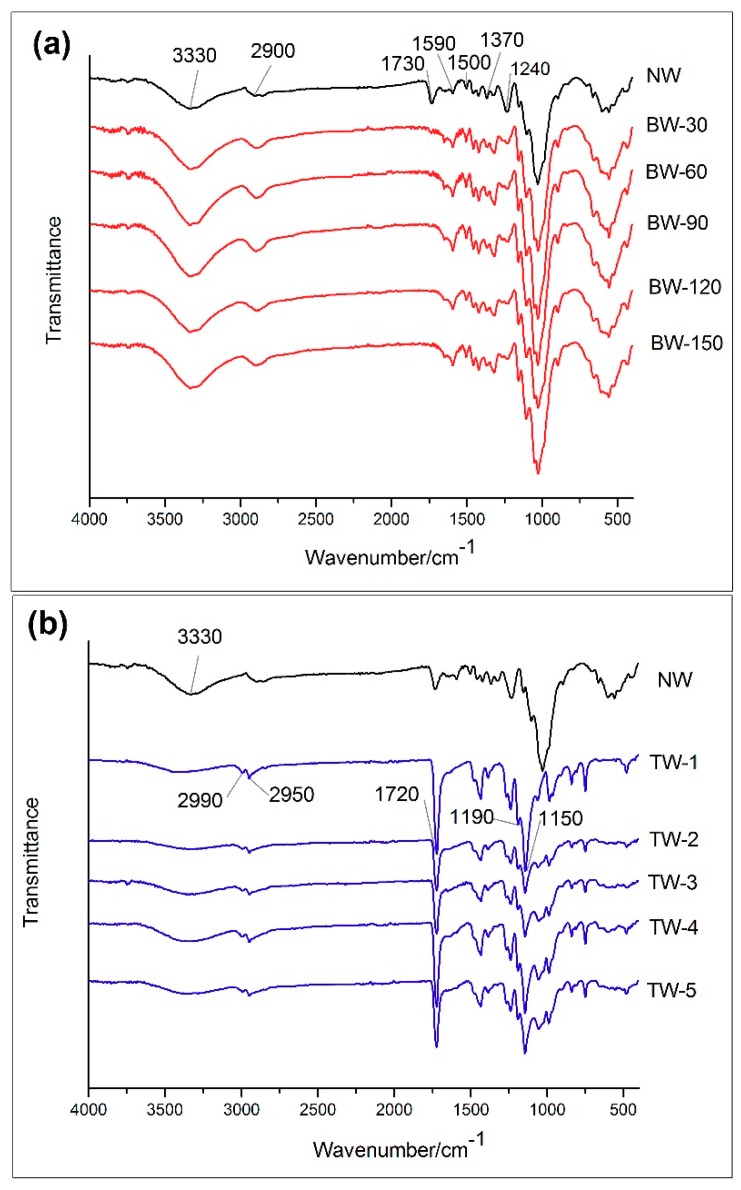
(**a**) FTIR spectra for natural wood (NW) and bleached wood (BW) samples. (**b**) FTIR spectra for natural wood (NW) and transparent wood (TW) samples.

**Figure 4 polymers-11-00776-f004:**
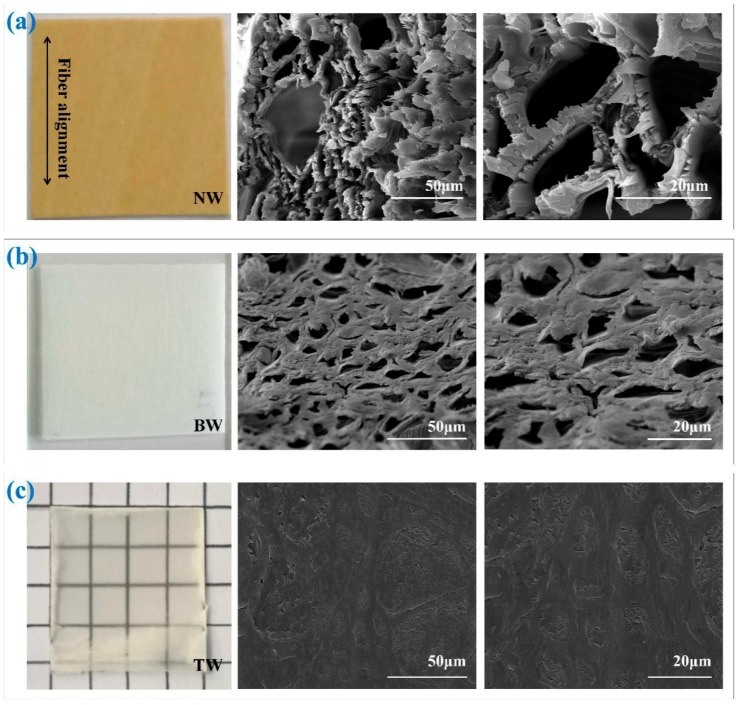
Photographs and SEM images of (**a**) natural wood (NW), (**b**) bleached wood (BW-150) and (**c**) transparent wood (TW-5) samples.

**Figure 5 polymers-11-00776-f005:**
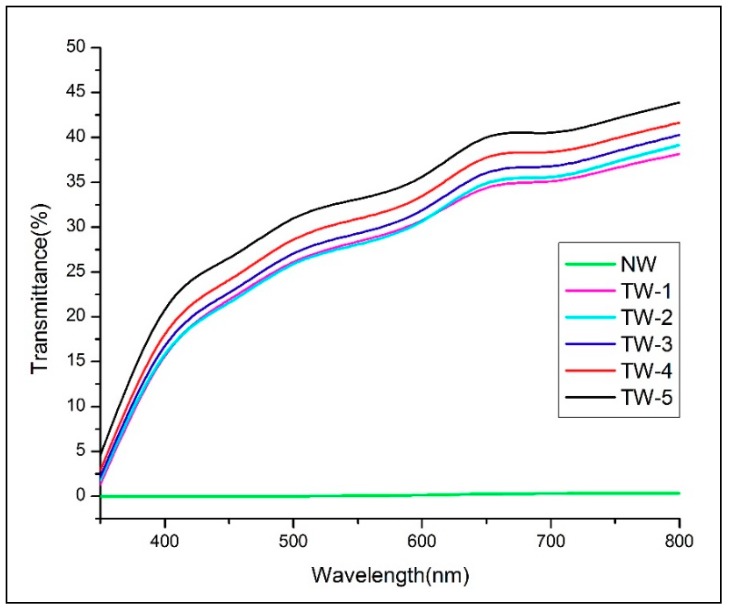
Optical transmittance of natural wood (NW) and transparent wood (TW) samples.

**Figure 6 polymers-11-00776-f006:**
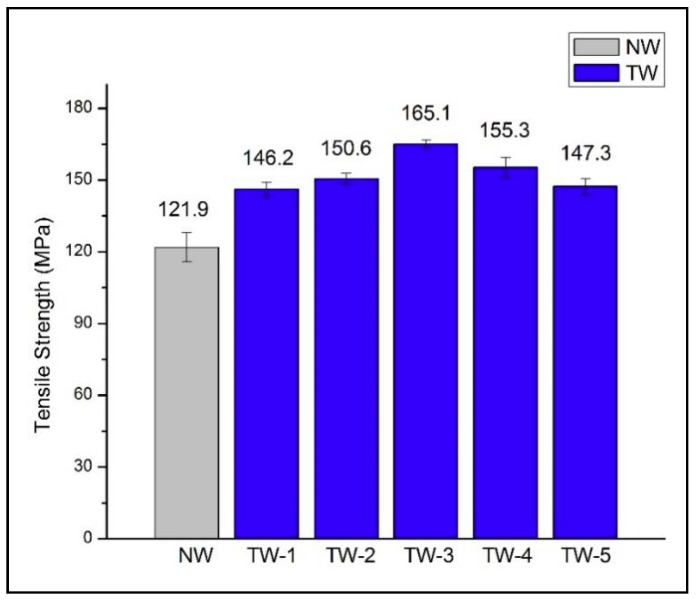
Tensile strength of natural wood (NW) and transparent wood (TW) samples.

**Figure 7 polymers-11-00776-f007:**
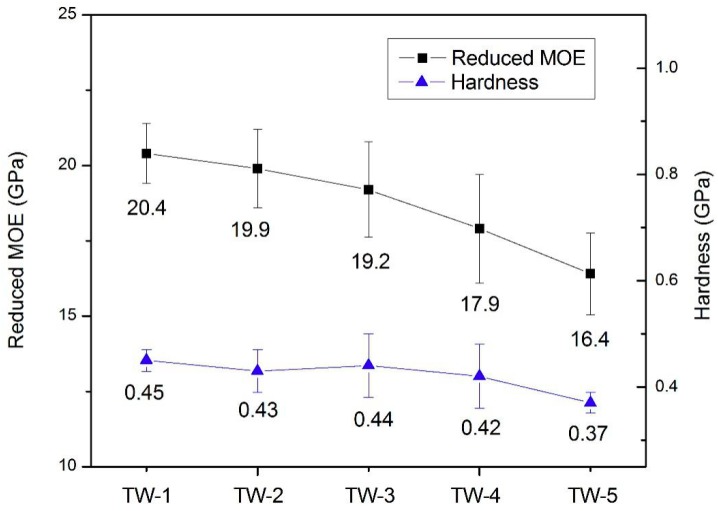
The reduced elastic modulus (MOE) (*E_r_*) and hardness (*H*) of transparent wood (TW) samples.

**Table 1 polymers-11-00776-t001:** Chemical composition content changes of various degrees of H_2_O_2_ bleaching treatment.

Sample Name	Cellulose Content (%)	Hemicellulose Content (%)	Lignin Content (%)
NW	48.3	17.2	24.3
BW-30	45.0	16.8	19.5
BW-60	44.4	16.5	18.4
BW-90	44.0	16.4	16.6
BW-120	43.9	15.9	15.2
BW-150	43.5	15.7	14.9

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
