# Peer review of "Effect of H2O2 Bleaching Treatment on the Properties of Finished Transparent Wood"

_polymers, 2019, doi:10.3390/polym11050776_

Reviewer 1 Report

Reviewers Comments:

Wu et al., fabricated transparent wood from environment-friendly hydrogen peroxide (H2O2) bleached basswood (Tilia) template using polymer impregnation. Author did very nice job. However, still so many issues should be resolved before its publication.

1.      Abstract should modify with some points regarding the characterization such as FTIR, FESEM and Tensile etc.

2.      Line 17,………….In spite of component removal, we did find the wood cell micro-structures were maintained during H2O2 bleaching treatment………….should recheck

3.      Abstract should be quantitative. Modify it.

4.      The tensile values needs standard deviation such as 28.50±12. Please check whole manuscript. If possible cite this paper: Carbohydrate Polymers, 211, 2019, 181-194.

5.      The novelty regarding some sentences should be insert in introduction part of manuscript. Previous years (especially 2018 & 2019) published should be insert with proper comparison with your work.

6.      The fabrication method needs references. Check it.

7.      Conclusion part should be rewrite with quantitatively. Check it.

8.       At page 10…Line 257…With the removal of the wood cell’s structural biopolymers at extended bleaching times, the interaction between the main components of the cell wall was weakened, which destroyed the integrity of the cell wall to a certain extent and decreased the Er of the cell wall……needs proper references.

9.      At page 9, line 247………This could be due to the wood templates at longer bleaching times being mechanically weaker than the templates produced at shorter bleaching times……..needs references.

10.  Equation 1,2, 3 and 4 needs citation. Check it and modify.

Author Response

I am pleased to resubmit for the revised version of manuscript entitled “Effect of H2O2 bleaching treatment on the properties of finished transparent wood”. Thank you for reading our manuscript and reviewing it. Those comments are all valuable and very helpful for revising and improving our paper. We have revised our manuscript carefully and have made correction which we hope meet with approval. So we have sent the revised manuscript and have highlighted changes by using the green colour. The main corrections in the paper and the responds to the reviewers’ comments are as following:

Responds to the reviewers’ comments:

Reviewer 1 Comments: Wu et al., fabricated transparent wood from environment-friendly hydrogen peroxide (H2O2) bleached basswood (Tilia) template using polymer impregnation. Author did very nice job. However, still so many issues should be resolved before its publication.

1. Abstract should modify with some points regarding the characterization such as FTIR, FESEM and Tensile etc.

2. Line 17,………….In spite of component removal, we did find the wood cell micro-structures were maintained during H2O2 bleaching treatment………….should recheck

3. Abstract should be quantitative. Modify it.

Response 1-3: The abstract has been modified.

“In spite of component removal, we did find the wood cell micro-structures were maintained during H2O2 bleaching treatment. This allowed for successful impregnation of polymer into the bleached wood template and strong transparent wood products. The transparent wood possessed optical transmittance up to 44 % at 800 nm with 14.7 % lignin content. Moreover, the transparent wood displayed an excellent tensile strength up to 165.1 MPa. The elastic modulus (Er) and hardness (H) of the transparent wood samples were lowered along with the increasement of H2O2 bleaching treatment time.” changed to Fourier transform infrared spectroscopy (FT-IR) and scanning electron microscope (SEM) analysis indicated the wood cell micro-structures were maintained during H2O2 bleaching treatment. This allowed for successful impregnation of polymer into the bleached wood template and strong transparent wood products. The transparent wood possessed a maximum optical transmittance up to 44 % at 800 nm with 150 min bleaching time. Moreover, the transparent wood displayed a maximum tensile strength up to 165.1 ± 1.5 MPa with 90 min bleaching time. The elastic modulus (Er) and hardness (H) of the transparent wood samples were lowered along with the increasement of H2O2 bleaching treatment time. And the transparent wood with 30 min bleaching time exhibited the highest Er and H values of 20.4 and 0.45 GPa. 

4. The tensile values needs standard deviation such as 28.50±12. Please check whole manuscript. If possible cite this paper: Carbohydrate Polymers, 211, 2019, 181-194.

Response: The tensile values has been modified and this paper:“Carbohydrate Polymers, 211, 2019, 181-194.”has been cited.

Figure 6 shows the comparison of tensile strength for NW and a series of TW samples. All TW samples exhibited stronger tensile strength than NW samples (121.9 MPa). These results demonstrated that not only the transparency was significantly increased by polymer infiltration, mechanical properties also be improved compared to NW samples [37]. The longitudinally oriented cellulose nanofiber structure and the interaction between wood cellulose nanofibers and PMMA were favorable to the strength of TW samples [38]. It was interesting to find that of all the TW samples, TW-3 exhibited the greatest tensile strength (165.1 MPa). ” changed to Figure 6 shows the comparison of tensile strength for NW and a series of TW samples. The tensile strength of TW samples were related to their compositions and chemical structures [39]. All TW samples exhibited stronger tensile strength than NW samples (121.9 ± 6.12 MPa). These results demonstrated that not only the transparency was significantly increased by polymer infiltration, mechanical properties also be improved compared to NW samples [40]. The longitudinally oriented cellulose nanofiber structure and the interaction between wood cellulose nanofibers and PMMA were favorable to the strength of TW samples [41]. It was interesting to find that of all the TW samples, TW-3 exhibited the greatest tensile strength (165.1 ± 1.5 MPa). ”

5. The novelty regarding some sentences should be insert in introduction part of manuscript. Previous years (especially 2018 & 2019) published should be insert with proper comparison with your work.

Response: Qiu et al. (2019) also used 1.5 wt % NaClO2 with acetate buffer solution (pH 4.6) as lignin removal solution to impregnate wood samples, and lignin content decreased from 23.5 ± 1.8 % for the untreated wood to 1.6 ± 0.2 % for the delignified wood after 8 h delignification [12]. Using a different approach, Zhu et al. (2016) reported a means of delignification using a boiling aqueous solution of NaOH and Na2SO3 to which additional H2O2 was added. In this case, lignin content in the template woods was less than 3% [13]Liu et al. (2018) also obtained delignified wood by using 5 g NaOH and 15 g Na2SO3 mixing 400 mL methanol (20 % volume fraction) water solution to extract wood samples, then the samples were placed in the 1.5 mol/L H2O2 solution until the wood yellow color disappeared and the removal rate of lignin reached up to 99.2 % [14].Generally, there are harmful components, such as methyl mercaptan, dimethyl sulfide, and hydrogen sulphide, generated during the delignification process [15]. ”has been added.

6. The fabrication method needs references. Check it.

Response: The references has been added.

“Natural wood (NW) samples were first dried at 103 ℃ for 24 h prior to further bleaching treatment. The bleaching solution was prepared through mixing 6 wt% H2O2, 1 wt% trisodium citrate dihydrate, 1 wt% NaOH, and 92 wt% ultrapure water [16].” and “To begin transparent wood production, pure MMA monomer was uniformly mixed with AIBN initiator (0.5 wt% solution) and pre-polymerized at 75 ℃ for 15 min [11]. ”

7. Conclusion part should be rewrite with quantitatively. Check it.

Response: The conclusion part has been modified. An environmentally friendly H2O2 bleaching treatment method was successfully taken to make TW samples. FT-IR and SEM observations showed that the microstructure of the wood cells was still well preserved after bleaching treatment and the MMA was successfully impregnated into the wood template. The optical transmittance and tensile strength of TW samples produced from these templates was superior to NW samples, in spite of the severe chemical compositional changes imparted by the bleaching treatment. The transparent wood possessed a maximum optical transmittance up to 44 % at 800 nm and displayed a maximum tensile strength up to 165.1 ± 1.5 MPa. The Er and H of the cell wall templates present in the TW sample was also found to be related to the removal of chemical components, as they both decreased at increasing bleaching times. And they exhibited the highest Er and H values of 20.4 and 0.45 GPa.

8. At page 10…Line 257…With the removal of the wood cell’s structural biopolymers at extended bleaching times, the interaction between the main components of the cell wall was weakened, which destroyed the integrity of the cell wall to a certain extent and decreased the Er of the cell wall……needs proper references.

Response: The references has been added. “with the removal of the wood cell’s structural biopolymers at extended bleaching times, the interaction between the main components of the cell wall was weakened, which destroyed the integrity of the cell wall to a certain extent and decreased the Er of the cell wall [43]. ”

9. At page 9, line 247………This could be due to the wood templates at longer bleaching times being mechanically weaker than the templates produced at shorter bleaching times……..needs references.

Response: The references has been added. “This could be due to the wood templates at longer bleaching times being mechanically weaker than the templates produced at shorter bleaching times[42].”

10. Equation 1,2, 3 and 4 needs citation. Check it and modify.

Response: The citation has been added.

“The nanoindentation test method, the reduced elastic modulus (MOE) (Er) and hardness (H) measurements upon TW samples, and calculation methods (formula (3) and (4))were all performed according to our previously described research methods [20-21]. ” and “ In addition, the content and removing rate of the chemical composition in Table 1 and Figure 2 can be determined by formula 1 and 2, respectively [37]. ”

We appreciate for Editor and Reviewers’ warm work earnestly, and hope that the correction will meet with approval. Once again, thank you very much for your comments and suggestions.

Yours sincerely,

Yan Wu

Reviewer 2 Report

Reviewer comments:

Title:  Effect of H2O2 bleaching treatment on the properties 3 of finished transparent wood

Dear Authors,

Authors deal with interesting and actual.  Manuscript is well written and good literature overview was made.   There are used appropriate analytical methods only some additional description some methods should be add.  Below can be found my comments, which I hope that will improve manuscript.

Introduction:

Line 53:  Hu et al. (2016) does not matched with reference no. 12 (line 55)

Materials and methods:

Line 115 -127: Sections 2.4-2.5 To this sections add information (dimensions, replicates, …) about samples which was tested (observed) according to different methods.

Results and discussion:

Figures have to be self-readable so please to caption add abbreviation write with full names.  

Line 174: Caption of Figure 3 should be changed. Add information what mean NW, TW and BW and move latter of graphs behind the description.

Line 194: Caption of Figure 4 should be changed. Add information what mean NW, TW and BW  

Line 199: “Figure 4b) was colorless relative to the NW sample« I see this sample more white then colour less. I suggest to change the Figure where will be also mesh grid beneath like on Figure 4c.  

Line 223: Caption of figure 5 should be changed. Add information what mean NW and TW

Line 240: See previous comment for figure 5

Line 251: See previous comment for figure 5

Author Response

I am pleased to resubmit for the revised version of manuscript entitled “Effect of H2O2 bleaching treatment on the properties of finished transparent wood”. Thank you for reading our manuscript and reviewing it. Those comments are all valuable and very helpful for revising and improving our paper. We have revised our manuscript carefully and have made correction which we hope meet with approval. So we have sent the revised manuscript and have highlighted changes by using the green colour. The main corrections in the paper and the responds to the reviewers’ comments are as following:

Responds to the reviewers’ comments:

Reviewer 2 Comments:Authors deal with interesting and actual. Manuscript is well written and good literature overview was made.There are used appropriate analytical methods only some additional description some methods should be add.Below can be found my comments, which I hope that will improve manuscript.

1. Introduction:Line 53:  Hu et al. (2016) does not matched with reference no. 12 (line 55)

Response: The reference has been modified.

Using a different approach, Zhu et al. (2016) reported a means of delignification using a boiling aqueous solution of NaOH and Na2SO3 to which additional H2O2 was added. In this case, lignin content in the template woods was less than 3% [13

The reference: 13. Zhu, M.; Song, J.; Li, T.; Gong, A.; Wang, Y.; Dai, J.; Yao, Y.; Luo W.; Henderson D.; Hu, L. Highly anisotropic, highly transparent wood composites. Adv. Mater. 2016, 28, 5181-5187. doi:10.1002/adma.201600427.

2. Materials and methods: Line 115 -127: Sections 2.4-2.5 To this sections add information (dimensions, replicates, …) about samples which was tested (observed) according to different methods.

Response: The information has been added.

2.4. Fourier Transform Infrared Analysis

The attenuated total reflection (ATR) Fourier transform infrared spectroscopy (FT-IR) spectra of NW, BW and TW samples were analyzed using a VERTEX 80V spectrometer. Spectra were collected over the range from 400 cm-1 to 4000 cm-1 by 16 scans at a resolution of 4 cm-1The dimensions of samples were 20 mm long × 20 mm wide × 0.4 mm thick and all samples were dried before analysis.

2.5. Scanning Electron Microscopy

The dried NW, BW and TW samples were coated with gold particles, and then observed with a FEI Quanta 200 scanning electron microscope (SEM) at an accelerating voltage of 20 kV. The cross sections of samples that perpendicular to the direction of the wood fiber alignment were observed.

3. Results and discussion:

Figures have to be self-readable so please to caption add abbreviation write with full names.

Line 174: Caption of Figure 3 should be changed. Add information what mean NW, TW and BW and move latter of graphs behind the description.

Line 194: Caption of Figure 4 should be changed. Add information what mean NW, TW and BW  

Line 199: “Figure 4b) was colorless relative to the NW sample« I see this sample more white then colour less. I suggest to change the Figure where will be also mesh grid beneath like on Figure 4c.  

Line 223: Caption of figure 5 should be changed. Add information what mean NW and TW

Line 240: See previous comment for figure 5

Line 251: See previous comment for figure 5

Response: The information has been added.

Figure 3. (a) FTIR spectra for natural wood (NW) and bleached wood (BW) samples. (b) FTIR spectra for natural wood (NW) and transparent wood (TW) samples.”

Figure 4. Photographs and SEM images of (a) natural wood (NW), (b) bleached wood (BW-150) and (c) transparent wood (TW-5) samples. ”

Figure 5. Optical transmittance of natural wood (NW) and transparent wood (TW) samples.”

Figure 6. Tensile strength of natural wood (NW) and transparent wood (TW) samples.”

Figure 7. The reduced elastic modulus (MOE) (Er) and hardness (H) of transparent wood (TW) samples.

We appreciate for Editor and Reviewers’ warm work earnestly, and hope that the correction will meet with approval. Once again, thank you very much for your comments and suggestions.

Yours sincerely,

Yan Wu

Reviewer 3 Report

Some editing for English spelling and style is needed.

Author Response

I am pleased to resubmit for the revised version of manuscript entitled “Effect of H2O2 bleaching treatment on the properties of finished transparent wood”. Thank you for reading our manuscript and reviewing it. Those comments are all valuable and very helpful for revising and improving our paper. We have revised our manuscript carefully and have made correction which we hope meet with approval. So we have sent the revised manuscript and have highlighted changes by using the green colour. The main corrections in the paper and the responds to the reviewers’ comments are as following:

Responds to the reviewers’ comments:

Reviewer 3 Comments:

We appreciate for Editor and Reviewers’ warm work earnestly, and hope that the correction will meet with approval. Once again, thank you very much for your comments and suggestions.

Yours sincerely,

Yan Wu

Reviewer 4 Report

This manuscript aims to study the manufacture of transparent wood using an environment-friendly hydrogen peroxide (H2O2) bleaching process basswood (Tilia) and impregnation with MMA polymer.Although the first studies were carried out in 1992, the transparent wood has attracted increasing research interests.

The state of the art includes the relevant studies that have been published about the bleaching of wood for to produce transparent wood. However, the applications of this new material should be indicated. The discussion about the properties would have more sense if it was focused the application in a final product. 

In the materials and methods section, several analytical methods were used not only to the assess the effect of bleaching in wood chemical composition and structure, but also to evaluate the performance of this product. However, additional information should be given as the methods used for the determination of cellulose, hemicellulose and lignin contents. In case of mechanical tests, the samples size as well as the test method should be provided. In the description of SEM analysis, it is not indicated the direction or plan of SEM images in relation to wood samples. Furthermore, it was not indicated the orientation of the samples and how they were cut, by rotary cutting or slicing of the log ? The fibre direction are parallel to the length direction ?? 

The results are well discussed. However, some conclusions are not supported by the results, as the SEM images that do not show that “the bleaching induced the formation of pits on the fiber, which is known to increase the surface area”.

In conclusion, I recommend the acceptance of the paper with minor revisions. 

Author Response

I am pleased to resubmit for the revised version of manuscript entitled “Effect of H2O2 bleaching treatment on the properties of finished transparent wood”. Thank you for reading our manuscript and reviewing it. Those comments are all valuable and very helpful for revising and improving our paper. We have revised our manuscript carefully and have made correction which we hope meet with approval. So we have sent the revised manuscript and have highlighted changes by using the green colour. The main corrections in the paper and the responds to the reviewers’ comments are as following:

Responds to the reviewers’ comments:

Reviewer 4 Comments:

This manuscript aims to study the manufacture of transparent wood using an environment-friendly hydrogen peroxide (H2O2) bleaching process basswood (Tilia) and impregnation with MMA polymer.Although the first studies were carried out in 1992, the transparent wood has attracted increasing research interests.

1. The state of the art includes the relevant studies that have been published about the bleaching of wood for to produce transparent wood. However, the applications of this new material should be indicated. The discussion about the properties would have more sense if it was focused the application in a final product.

Response: The applications were added in conclusion part. “The fingdings will provide knowledge base for the further application of transparent wood as novel home materials in interior decoration fields.”

2. In the materials and methods section, several analytical methods were used not only to the assess the effect of bleaching in wood chemical composition and structure, but also to evaluate the performance of this product.

① However, additional information should be given as the methods used for the determination of cellulose, hemicellulose and lignin contents.

In case of mechanical tests, the samples size as well as the test method should be provided.

In the description of SEM analysis, it is not indicated the direction or plan of SEM images in relation to wood samples. Furthermore, it was not indicated the orientation of the samples and how they were cut, by rotary cutting or slicing of the log ? The fibre direction are parallel to the length direction ?? 

Response:The information has been added.

“The cellulose, hemicellulose and lignin contents (including acid-insoluble lignin plus acid-soluble lignin) of NW and BW samples were tested by the Laboratory Analytical Procedure (LAP) written by the National Renewable Energy Laboratory (Determination of Structural Carbohydrates and Lignin in Biomass) . The main methods were as follows [19]: First, samples were processed into 20-80 meshes of wood powder and dried at 105℃ for 24 h. Then 0.3 ± 0.01 g dried wood powder were put into the hydrolytic flask. The 3.00 ± 0.01 ml 72 wt% concentrated sulfuric acid was added to the hydrolytic flask, and all wood powder samples were infiltrated at the same time. The hydrolysate flasks were all covered and placed in water bath at 30 ℃ for 1 h. Then the 84.00 ml ± 0.04 ml water was added in the flask. Then flasks were placed into the sterilizer at 121 ℃ for 1 h and opened after cooling. The hydrolyzed samples were filtered by a constant weight G3 funnel, and 50 ml filtrate was retained for the determination of acid-soluble lignin (measured within 6 hours) and sugar concentration. The filtered residue was rinsed with hot deionized water to neutral, and then placed in an oven at 105 ℃ until constant weight, and the weight was recorded. After constant weight, it was transferred to the muffle furnace and dried at 575 ± 25 ℃ for 24 ± 6 h. Later, the G3 funnel was taked out and the weight was recorded. The filtrate was diluted with the corresponding multiple, and the UV absorption value was determined by UV spectrophotometer at 205 nm. The filtrate was diluted by a certain multiple, and then the sugar content was analyzed by High performance liquid chromatography (HPLC).”

② 2.7. Mechanical Properties

The tensile strength of NW and TW samples were performed in a SANS-CMT6104 electromechanical universal testing machine. The tensile speed was set at 2 mm/min. The dimensions of samples were 20 mm long × 20 mm wide × 0.4 mm thick and stretched in the direction of the fiber alignment. “

2.1. Materials

Basswood (Tilia) with dimensions of 20 mm long × 20 mm wide × 0.4 mm thick (the depth of the lumina is as long as the length of the wood samplesand ultrapure water were supplied by Yihua Lifestyle Technology Co., Ltd., China. 

2.5. Scanning Electron Microscopy

The dried NW, BW and TW samples were coated with gold particles, and then observed with a FEI Quanta 200 scanning electron microscope (SEM) at an accelerating voltage of 20 kV. The cross sections of samples that perpendicular to the direction of the wood fiber alignment were observed.

3.The results are well discussed. However, some conclusions are not supported by the results, as the SEM images that do not show that “the bleaching induced the formation of pits on the fiber, which is known to increase the surface area”.

Response:“ In addition, bleaching induced the formation of pits on the fiber, which is known to increase the surface area.”has been deleted.

We appreciate for Editor and Reviewers’ warm work earnestly, and hope that the correction will meet with approval. Once again, thank you very much for your comments and suggestions.

Yours sincerely,

Yan Wu

Round  2

Reviewer 1 Report

Author has been addressed the recommended corrections. So I allow it for publication.